# Thin lamellar films with enhanced mechanical properties for durable radiative cooling

Lianhu Xiong[1], Yun Wei[1], Chuanliang Chen[1], Xin Chen[1], Qiang Fu ⓘ[1] ✉ & Hua Deng ⓘ[1] ✉

Passive daytime radiative cooling is a promising path to tackle energy, environment and security issues originated from global warming. However, the contradiction between desired high solar reflectivity and necessary applicable performance is a major limitation at this stage. Herein, we demonstrate a "Solvent exchange-Reprotonation" processing strategy to fabricate a lamellar structure integrating aramid nanofibers with core-shell $TiO_2$-coated Mica microplatelets for enhanced strength and durability without compromising optical performance. Such approach enables a slow but complete two-step protonation transition and the formation of three-dimensional dendritic networks with strong fibrillar joints, where overloaded scatterers are stably grasped and anchored in alignment, thereby resulting in a high strength of ~112 MPa as well as excellent environmental durability including ultraviolet aging, high temperature, scratches, etc. Notably, the strong backward scattering excited by multiple core-shell and shell-air interfaces guarantees a balanced reflectivity (~92%) and thickness (~25 μm), which is further revealed by outdoor tests where attainable subambient temperature drops are ~3.35 °C for daytime and ~6.11 °C for nighttime. Consequently, both the cooling capacity and comprehensive outdoor-services performance, greatly push radiative cooling towards real-world applications.

While working to limit global warming to 1.5 °C proposed in the Paris Agreement[1], it is undeniable that climate extremes around the world have been detected frequently, especially in hot summers, which has raised a series of energy, environmental, and security issues, including electricity consumption (e.g., air conditioners), carbon emissions, and self-ignition (e.g., vehicles and cables)[2,3]. Passive daytime radiative cooling (PDRC) is an energy-efficient and eco-friendly cooling technology, with fundamental principles of reflecting most sunlight (0.3–2.5 μm) and emitting long-wave infrared (LWIR) radiation through the atmospheric window (8–13 μm), has been highly regarded in recent years[4-8]. Ideally, sub-ambient temperature drop of objects can be passively reached with zero energy input and zero pollution output since there is a sustained cold universe (~2.7 K) as a power source for PDRC[4,9-11].

The key-point of PDRC materials is ultra-high reflectivity ($\bar{R}_{solar}$ > 90%) across 0.3–2.5 μm wavelengths[12,13], since just a few percent of solar absorbance would eclipse cooling capacity of LWIR radiation and effectively heat any exposed surfaces (Supplementary Fig. 1). Research in recent decades mainly focuses on two fields to design high-reflectivity PDRC films comprising porous structures[14-20] and polymer-dielectric scatterers composites[13,21-24]. However, the above-mentioned strategies, while achieving high reflectivity, typically mean losses of other applicable performance (e.g., strength, modulus, durability, and thickness), which makes PDRC films unable to meet the

[1]College of Polymer Science and Engineering, State Key Laboratory of Polymer Materials Engineering, Sichuan University, 610065 Chengdu, China. ✉e-mail: qiangfu@scu.edu.cn; huadeng@scu.edu.cn

long-term cooling requirements of outdoor devices in hot summers. Specifically, for porous structures, although air voids in polymer skeleton create strong multiple scattering, thus greatly enhancing solar reflectivity[14,15]. But increased cost and inherent mechanical weakness triggered by elevated thickness with abundant air holes or defects still remain as major problems, especially for all-polymeric PDRC films[12,25], which were seldom resistant to environmental aging (e.g., ultraviolet (UV) radiation, water rinsing, and scratch damage) and fire[26]. As for the improvement of solar reflectivity, introducing high-content dielectric scatterers to polymer matrix is another common alternative. Nevertheless, with overloaded scatterers (>50 wt%), both limited-processing and scatterers leakage has resulted in difficult shaping and applicability of PDRC materials[27,28]. More importantly, the severe agglomeration of dielectric particles directly leads to the substantial decline of optical and mechanical properties for PDRC stuffs[29], which was rarely discussed in previous reports. Therefore, pushing PDRC towards practical applications, alleviating the agglomeration of scatterers, and simultaneously realizing highly efficient PDRC and other applicable performances at relatively low cost remain as challenges.

Herein, from the perspective of an interfacial network, we demonstrated a hierarchical-morphology design mechanism, which incorporated dielectric scatterers to the porous structure, directly providing efficient PDRC and excellent comprehensive applicable performance. Two-dimensional (2D) core-shell scatterers composed of exfoliated Mica with surface uniformly distributed $TiO_2$ nanograins (Mica@$TiO_2$) were assembled with aramid nanofibers (ANFs) via a "Solvent exchange-Reprotonation" processing strategy, successfully forming a well-organized lamellar PDRC film (AMTA). Specifically, by regulating the slow but complete two-step protonation transition, a three-dimensional (3D) dendritic ANFs network with strong fibrillar joints is formed, where more than 50 wt% scatterers are stably grasped and orderly embedded in AMTA with the help of "hyperbranched ANFs

adhesives", thereby exhibiting excellent mechanical strength of ~112 MPa and Young's modulus of ~4 GPa. Besides, the barely agglomerated $TiO_2$ nanograins allow the sunlight to fully scatter at core-shell and shell-air interfaces, while the intense group vibrations in Mica greatly enhance the infrared absorption of AMTA within the atmospheric window, so as to achieve a high $\bar{R}_{solar}$ of 92% and acceptable $\bar{\varepsilon}_{LWIR}$ of 87% at a fairly low thickness of 25 μm. We experimentally validated the high-performance daytime cooling of AMTA with an average sub-ambient temperature drop of ~3.35 °C under direct sunlight, potentially applied to mobile phones, vehicles, and buildings for effective thermal management. Notably, our AMTA exhibited impressive environmental durability with its optical and mechanical properties retained, even after being challenged against 180 °C thermal treatment, 96 h UV radiation, 8 h water rinsing and severe scratch damage, promising for the real-world applications of PDRC.

## Results

### Design, fabrication, and characterization

To meet the long-term cooling requirements of outdoor devices in harsh environments, high solar reflectivity, excellent mechanical properties, and robust environmental durability of radiative coolers are the primary considerations. For solar reflectivity, favorable strong scattering created by impedance mismatch[30] at multiple interfaces has developed as its classical design principle. Mica@$TiO_2$, a commercial pearlescent pigment, creatively combines exfoliated Mica microplatelet (Core) with uniformly distributed $TiO_2$ nanograins (Shell, Supplementary Fig. 3) to achieve both high refractive index ($k > 2.6$) and environmental durability, making it a desirable 2D inorganic scatterer (Fig. 1b, c). Besides, functional groups such as Si-C, Si-O, and Ti-O within Mica@$TiO_2$ can greatly improve the emissivity of radiative coolers at the atmospheric window. Likewise, since the importance of polymer networks for the molding and processing of PDRC films, ANFs

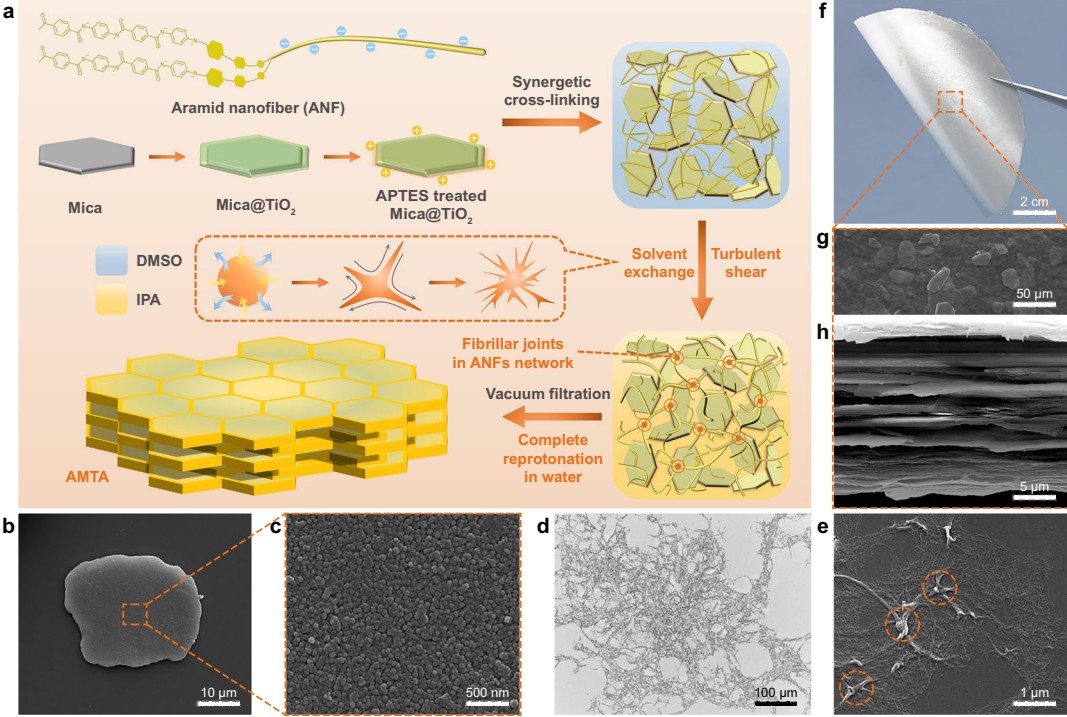

**Fig. 1 | Illustration of the fabrication process and microstructure characterizations. a** Specific "Solvent exchange-reprotonation" processing strategy. **b** FESEM image of Mica@$TiO_2$ microplatelet with lateral size of about 20 μm. **c** FESEM image of $TiO_2$ nanograins with diameters of about 30 nm uniformly distributed on Mica surface. **d**, **e** OM (**d**) and FESEM (**e**) images of ANFs network with fibrillar joints, showing hyperbranched morphology. **f** Photograph of AMTA (with 50 wt% Mica@$TiO_2$ loading) displaying its flexibility. **g**, **h** Top-view (**g**) and section-view (**h**) FESEM images of AMTA, showing a well-organized lamellar structure with interlayer micropores.

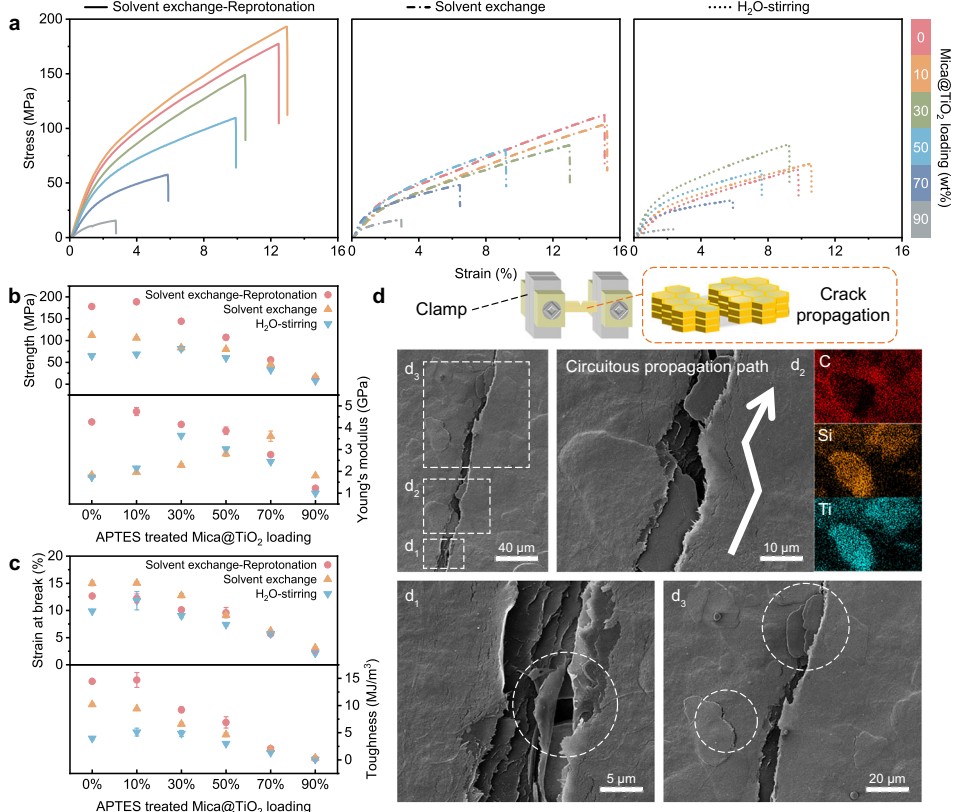

**Fig. 2 | Mechanical properties and multiscale extrinsic toughening mechanisms. a** Tensile stress-strain curves of ANFs/Mica@TiO$_2$ composites via different preparation methods. **b, c** Ultimate strength, maximum Young's modulus (**b**) and strain at break, work of toughness (**c**) obtained by uniaxial tensile tests. **d** Crack propagation behaviors of AMTA observed from FESEM image and EDS elemental maps of C, Si, and Ti, showing long-range crack deflection and branching, as well as strong plastic deformation and curling of dendritic ANFs adhesives. Source data are provided as a Source Data file.

with large specific areas, ultra-high strength (~3.6 GPa), and heat-resistance (>400 °C)[31] are appropriately used as an organic matrix to fabricate materials with high mechanical performance.

The lamellar PDRC films were assembled in a "Solvent exchange-Reprotonation" processing strategy schematically illustrated in Fig. 1a. Aramid microfibers (AMFs) were first dissociated into negatively charged ANFs by deprotonation in a mixture of dimethyl sulfoxide (DMSO) and potassium hydroxide (KOH). Mica@TiO$_2$ was pretreated with (3-aminopropyl)triethoxysilane (APTES) to make it positively charged (Supplementary Fig. 5). Afterward, ANFs were mixed with APTES-treated Mica@TiO$_2$, forming a homogeneous flaxen sol via synergetic cross-linking such as electrostatic interaction and hydrogen bonding. Subsequently, the uniform sol was slowly dropped into the high-velocity flow field of isopropanol (IPA), involving solvent exchange and turbulent shear[32]. During the process of preliminary protonation, we note that ANFs flocculated into hyperbranched morphology, which was conducive to assembling highly connected networks with fibrillar joints (Fig. 1d, e), so that high-content Mica@TiO$_2$ could be stably dispersed for several hours (Supplementary Fig. 7). Moreover, the vacuum-assisted filtration (VAF) method was used to prepare lamellar films with interlayer micropores (Fig. 1h) based on the above dendritic colloid suspension. These films were further immersed into a water bath to reprotonate the ANFs network completely, resulting in high mechanical performance radiative coolers (Fig. 1f).

**Mechanical performance and toughening mechanisms analysis**
Mechanical performance is particularly considerable for devices that work outdoors for long periods of time. To demonstrate the superiority of the above "Solvent exchange-Reprotonation" processing strategy, the mechanical properties of lamellar films were systematically studied and compared to those composites fabricated through conventional methods[31,33]. As shown in Fig. 2a–c, the tensile strength, Young's modulus as well as toughness of H$_2$O-stirring composites with the optimal Mica@TiO$_2$ loading (30 wt%) can barely reach ~81.5 MPa, ~3.6 GPa and ~4.9 MJ/m$^3$, respectively. As the ANFs overquick reprotonation provided by the strong proton donor (H$_2$O) is uneven and incomplete. Instead by solvent exchange and turbulent shear in IPA, a slow protonation process and the formation of a dendritic ANFs network with fibrillar joints can significantly improve the mechanical properties of the composites. Furthermore, through the complete reprotonation in water, the robust highly ordered lamellar microstructure is achieved with the strong electrostatic interaction and multiple hydrogen bonding between microfibers, demonstrating a tensile strength of ~193.2 MPa, a Young's modulus of 4.9 GPa and a toughness of 16.2 MJ/m$^3$, which are far superior to the previously reported PDRC films. Nevertheless, excessive Mica@TiO$_2$ will produce undesirable ultra micropore defects[34], leading to an obvious degradation in the mechanical properties of composites. Despite that, the strength of AMTA over 100 MPa has apparently satisfied the practical applications of efficient PDRC.

To further analyze the hierarchical toughening mechanisms of these ANFs/Mica@TiO$_2$ lamellar films at several scales, the mechanical response including crack propagation behaviors are investigated by SENT experiments and corresponding finite-element simulations, as illustrated in Fig. 2d and Supplementary Figs. 11 and 12. The major crack starts from the notch and propagates along a circuitous path within the lamellar films. Specifically, the soft and rigid two-component network provides diverse options for stress transfer and crack extension, which allows the crack to deflect and branch (Fig. 2d

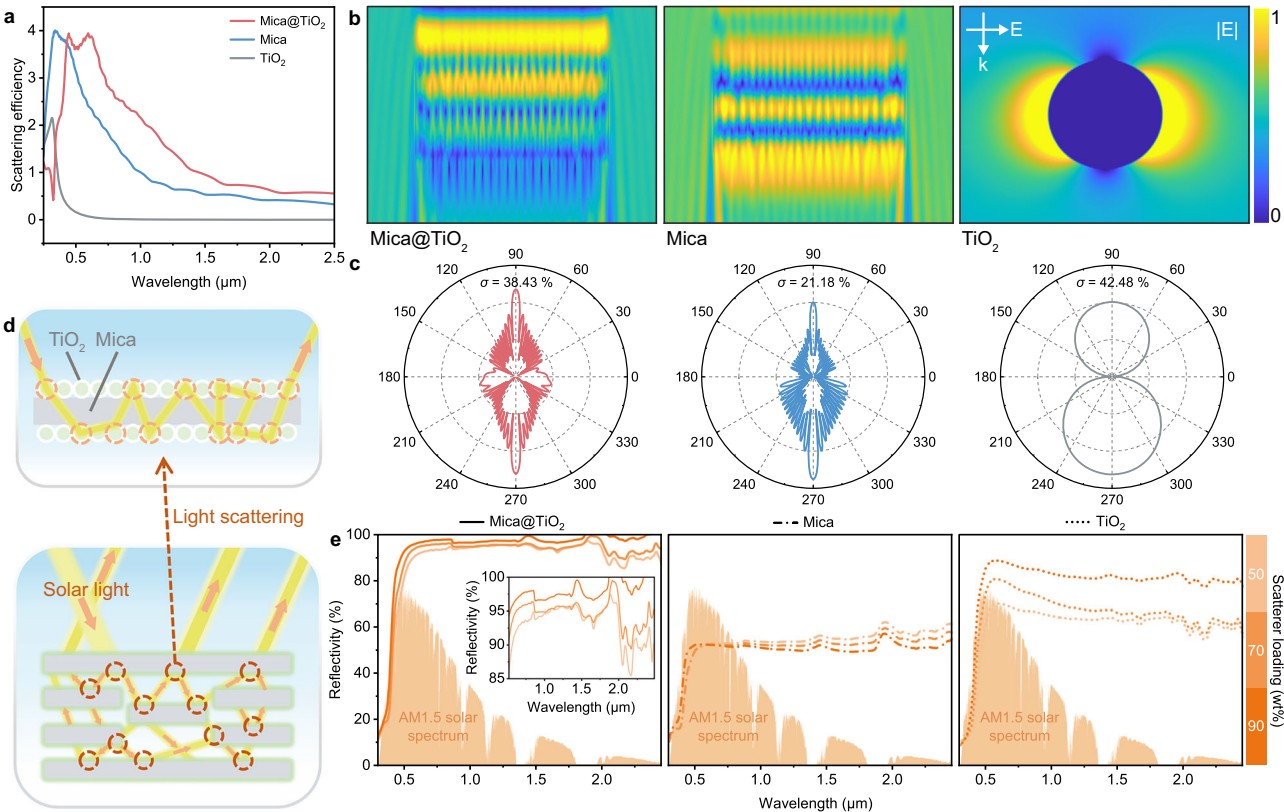

**Fig. 3 | Optical properties comparison among Mica@TiO₂, Mica, and TiO₂.**
**a**–**c** Simulated scattering efficiencies (**a**), near-field electric field distributions at 0.5 μm wavelength (**b**), and far-field scattering phase function at 0.5 μm wavelength (**c**) for different monodisperse scatterers, where the size consistent with real situation. E and k represent the electric field and wave vector of the incident light, respectively. 90° denotes as incident direction. **d** Schematic illustration of multiple scattering behaviors of AMTA caused by dielectric contrast, including Mica@TiO₂ and interlaminar micropores. **e** Solar reflectivity spectra of PDRC films respectively correspond to Mica@TiO₂, Mica, and TiO₂ scatterers in the UV–VIS–NIR region, AM 1.5 global solar spectrum was shaded as reference. Inset highlights the reflectivity spectra of PDRC films with Mica@TiO₂ scatterers. Source data are provided as a Source Data file.

and Supplementary Fig. 11e, j) when the embedded Mica@TiO₂ microplatelets are encountered, resulting in abundant tiny cracks uniformly distributed inside adjacent layers (Supplementary Fig. 11g). With increasing strain, Mica@TiO₂ that spans the whole major crack to withstand more stress[34,35] and branched cracks that bridged at the tip for toughening have also been observed (Supplementary Fig. 11h–j). Meanwhile, strong plastic deformation and curling of dendritic ANFs adhesives as well as pull-out and delamination of Mica@TiO₂ at the interface failure (Supplementary Fig. 13) are considered efficient energy dissipation modes[36–38] by frictional sliding and breaking of polymer networks, which are highly consistent with the results of theoretical simulations (Supplementary Fig. 12).

## Optical properties and theoretical radiative cooling performance

Ultra-high solar reflectivity is one of the necessary requirements for these PDRC films, which is mainly determined by the multiple scattering behaviors of sunlight inside materials. Regarding light scattering, it can be divided into three predominantly defined scattering regions according to the size of scatterers, namely, Rayleigh scattering, Mie scattering, and geometric scattering[39]. Because of the larger or comparable size of micro-sized Mica and nano-sized TiO₂ to the wavelength of sunlight, Mica@TiO₂ mainly follows the Mie scattering and geometric scattering mechanisms. Based on the scattering theory, the scattering efficiency ($Q_{sca}$) and scattering phase function play crucial roles in the final reflectivity. The $Q_{sca}$ is derived from the scattering cross-section ($C_{sca}$), which is determined by the ratio of scattering light to incident light, while the scattering phase function

denotes the intensity of scattering light in various directions. Since backscattering makes greater contributions to the final reflectivity, a parameter ($\sigma$) is redefined to describe the ratio of backscattering to total scattering.

$$\sigma = \frac{S_{back}}{S_{total}} \times 100\% \tag{1}$$

where $S_{back}$ is the integrated area of backscattering, and $S_{total}$ is the total integrated area of the scattering phase function.

Finite-difference time-domain (FDTD) simulations were employed to investigate the optical properties of different monodisperse scatterers. As shown in Fig. 3a, compared with Mica microplatelet or TiO₂ nanograin alone, core-shell Mica@TiO₂ exhibits higher $Q_{sca}$ in the VIS–NIR band and reaches the peak at ~500 nm where the solar irradiation is the strongest. That means that sunlight would diffuse intricate pathways, leading to an enormous increase in the proportion of reflected light. To further verify the backscattering behavior of Mica@TiO₂, near-field electric field distribution, and far-field scattering phase function were established (Fig. 3b, c). It can be seen that the scattering direction of TiO₂ has no obvious deviation owing to the Rayleigh scattering (Supplementary Movie 3), while the Mica and Mica@TiO₂, represent the overall forward scattering, which is one of the typical characteristics of Mie scattering. Nevertheless, in-depth analysis shows that Mica@TiO₂ is provided with a much stronger backward electric field distribution, which is different from Mica (Supplementary Movies 1 and 2). Meanwhile, in the far-field scattering phase function, Mica@TiO₂ is also consistent with expectation, i.e., the

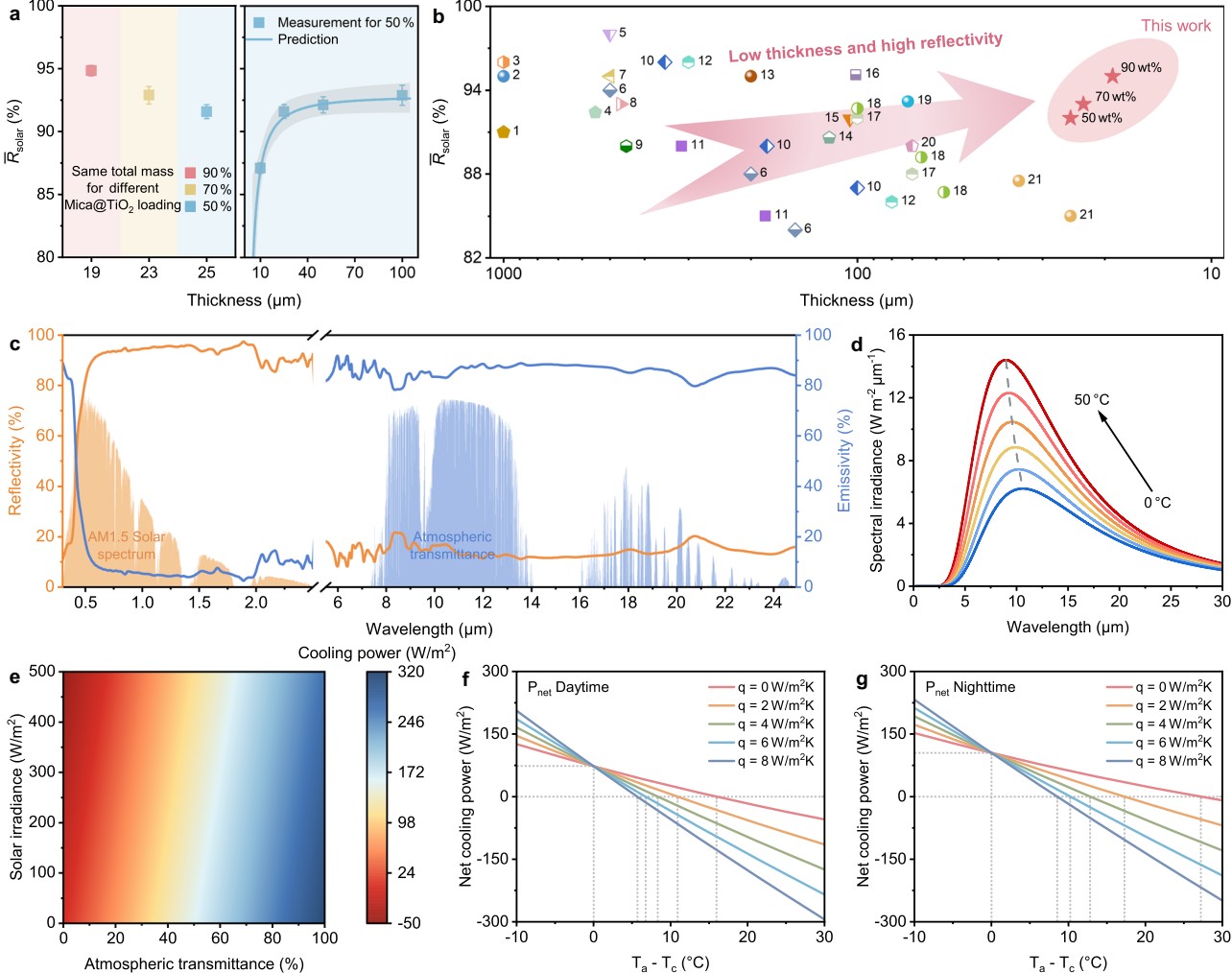

**Fig. 4 | Comprehensive optical properties and theoretical cooling performance. a** Thickness optimization of AMTA based on solar reflectivity. **b** Comparison of $\bar{R}_{solar}$ as a function of thickness with previous reports. **c** Spectral reflectivity and emissivity of a 25-μm-thick AMTA across 0.3–25 μm wavelength range against AM 1.5 solar spectrum and realistic atmospheric window. **d** Blackbody emissivity spectra from 0 to 50 °C, demonstrating that the emissivity of about

10 μm wavelength is more effective for improving the radiative cooling capacity of PDRC films. **e** Calculated cooling power of AMTA under different solar irradiance and atmospheric transmittance. **f**, **g** Calculated net cooling power of AMTA at daytime (**f**) and nighttime (**g**), respectively, with different $q$ values. Source data are provided as a Source Data file.

deflection angle of the scattering light to be as large as possible, preferably backward (larger $\sigma$).

Since just a few percent of solar absorbance would effectively heat the surface, the spectral reflectivity of the PDRC films across the ultraviolet–visible–near-infrared (UV–VIS–NIR) band is further quantitatively characterized (Fig. 3e). A 25 μm thick AMTA reflects more than 92% solar irradiation (0.3–2.5 μm), and increases rapidly with the elevation of Mica@TiO$_2$ loading. This is significantly superior to the TiO$_2$-based composites at the same mass fraction, which also corresponds to the previous optical simulations. The distribution of scatterers in AMTA interiorly is a key factor affecting its optical properties. Specifically, regular, dense, and non-agglomerated TiO$_2$ nanograins (Supplementary Figs. 3 and 4) wrapped on the Mica are conducive to creating abundant dielectric contrast interfaces for light scattering (Fig. 3d) owing to their high refractive index. So that AMTA is capable of achieving high reflectivity at fairly low thickness (≤25 μm), which is much thinner than the conventional polymeric PDRC films usually >100 μm (Fig. 4b). On the other hand, similar to the porous polymers reported in the past[15], the interlaminar micropores caused by the solvent evaporation contribute to the high reflectivity of AMTA as well through lamellar interface scattering (Supplementary Figs. 19 and 20).

It is worth noting that AMTA with internal Si-C, Si-O and Ti-O vibrations (Supplementary Fig. 8a) shows acceptable atmospheric window emissivity ($\bar{\varepsilon}_{LWIR}$ ~ 87%) at such a low thickness, which directly determines the theoretical upper limit of the cooling power. According to the heat transfer mode and the law of conservation of energy, combined with the measured full-wave band spectrum (Fig. 4c), the cooling power of AMTA can be theoretically calculated by the relevant equations (detailed calculation methods are shown in the Supplementary Information). Considering the diversity of solar irradiance ($I_{solar}$) and atmospheric transmittance ($t_{LWIR}$) at different longitude and latitudes in the world, we first calculated the cooling power of AMTA under various environments, as shown in Fig. 4e. As expected, lower $I_{solar}$ and higher $t_{LWIR}$ are beneficial for higher cooling power. And even in a relatively harsh environment (high $I_{solar}$ and low $t_{LWIR}$), ~100 W/m$^2$ cooling power can be obtained, demonstrating the universal and efficient radiative cooling capacity of AMTA. Moreover, the calculated net cooling power ($P_{net}$) and achievable cooling temperature ($T_a - T_c$) of AMTA for different non-radiative heat coefficients ($q$) are referred to in Fig. 4f, g. There are two key points that should be focused on: one is the $P_{net}$ at $T_a - T_c = 0$ ($P_{net}(T_a = T_c)$), which means that the cooling power totally derived from thermal radiation, and the other is the maximum

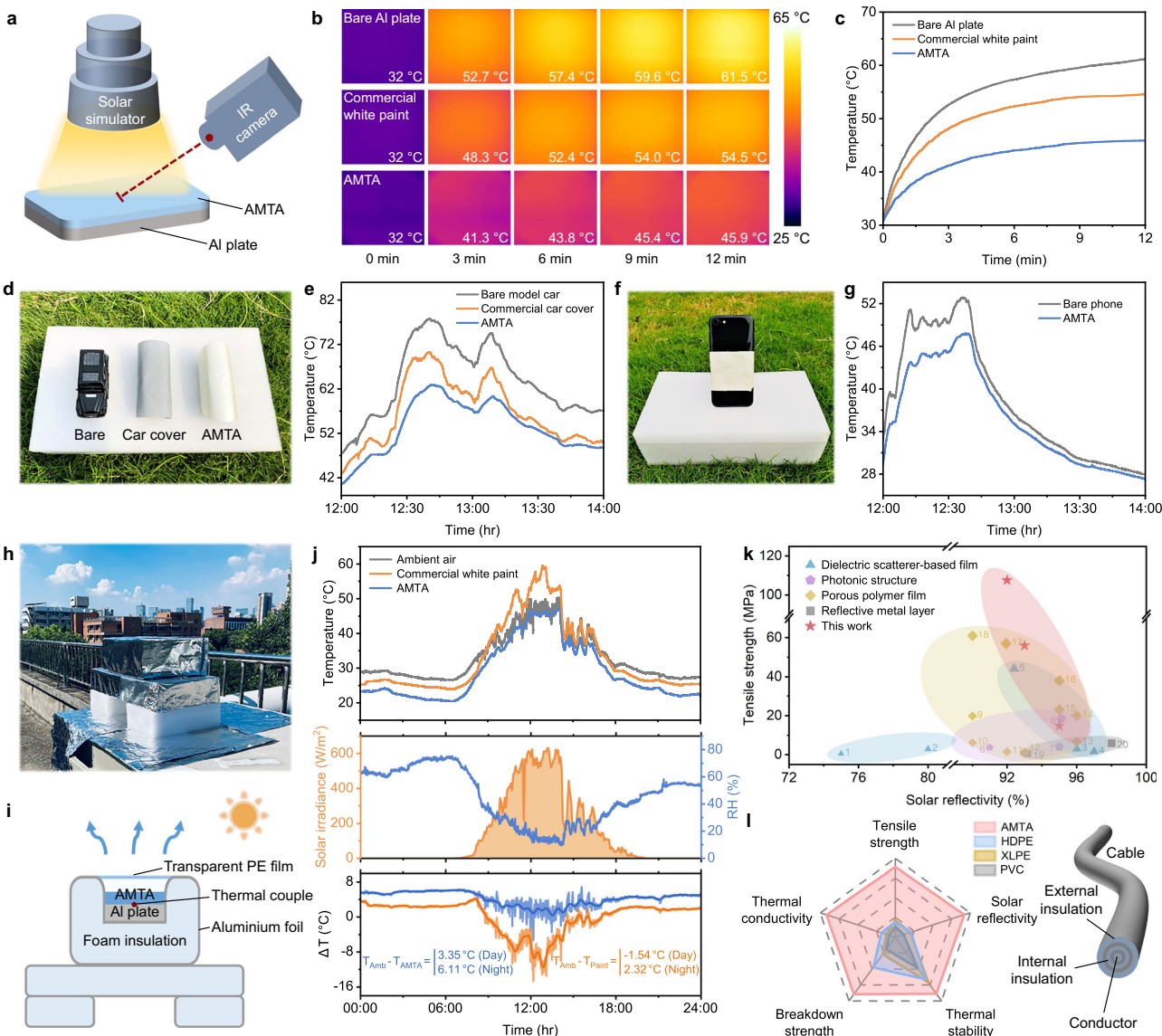

**Fig. 5 | Subambient radiative cooling performance and potential applications.**
**a** Schematic of a simulated solar heating test. **b**, **c** Thermal infrared imaging (**b**) and temperature tracking (**c**) of bare, commercial white paint-covered, and AMTA-covered Al plate, respectively. **d**, **e** Digital picture (**d**) and corresponding temperature variation (**e**) of the commercial car cover and AMTA on the surface of identical car models at high noon in summer. **f**, **g** Digital picture (**f**) and corresponding temperature variation (**g**) of the mobile phone covered with AMTA tailored for its shape and size, producing a comfortable state for outdoor running. **h**, **i** Photograph (**h**) and schematic (**i**) of home-built setup for sub-ambient radiative cooling measurements. **j** 24 h real-time temperature recordings of ambient air, commercial white paint-covered and AMTA-covered Al plate (August 13, 2022). Detailed solar irradiance ($I_{solar}$), relative humidity (RH), and sub-ambient temperature drop ($\Delta T$) were also shown in the corresponding graph. **k** Comparison of tensile strength and solar reflectivity of AMTA with various reported PDRC films. **l** Comprehensive performance comparison of AMTA with other universal cable insulation sheaths. Source data are provided as a Source Data file.

cooling temperature at $P_{net} = 0$ (($T_a - T_c$)$_{max}$). Comparing Fig. 4f with Fig. 4g, we can notice that $P_{net}(T_a = T_c)$ and ($T_a - T_c$)$_{max}$ are both lower during daytime. This is mainly results from the solar absorbance under direct sunlight since the $\bar{R}_{solar} < 100\%$. Notably, $P_{net}(T_a = T_c)$ of AMTA reaches 74 W/m$^2$ and 106 W/m$^2$ during daytime and nighttime, respectively, both resulting in more than 15 °C ($T_a - T_c$)$_{max}$ when there is zero non-radiative heat. More significantly, a large temperature drop of ~6 °C can be still achieved during daytime even with a $q = 8$ W/m$^2$K.

## Subambient radiative cooling performance and potential applications

In order to preliminarily evaluate the capability of sunlight reflection and thermal management of AMTA, simulated sunlight of 1000 W/m$^2$ was carried out to heat the aluminum (Al) plate, while infrared images and temperatures were captured simultaneously by an IR camera

(Fig. 5a–c). Within 12 min, the temperature of the AMTA-covered Al plate only rose from 32 to 46 °C and gradually leveled off, much lower than the bare Al plate (61 °C) and the one coated with commercial white paint (54 °C). This is mainly due to the effective solar heat shielding and considerable heat dissipation by AMTA (Supplementary Fig. 22). The actual sub-ambient radiative cooling performance of AMTA was characterized in Chengdu, China (104°2′35″E, 30°38′32″N, 570-m altitude) in early August employing the home-built contraption shown in Fig. 5h, i. For better steady and accuracy, insulated polystyrene foams and transparent polyethylene films were applied to eliminate the influence of thermal conduction and convection, respectively. The AMTA tightly covered on the Al plate was subjected to direct sunlight on a clear day, and its temperature variation throughout the day was monitored by a thermocouple. Additionally, temperatures of both the environment and the Al plate coated with

commercial white paint were also registered for comparison. The corresponding subambient temperature drop ($\Delta T$) can be obtained by subtracting the ambient temperature curves, as shown in Fig. 5j. Promisingly, the AMTA exhibited continuous high-performance subambient radiative cooling. Under the highest $I_{solar}$ of ~600 W/m² and a relative humidity (RH) of ~20% at noon, a critical average $\Delta T$ of 3–4 °C was achieved. At night, it can easily reach a high cooling effect of more than 6 °C even under an RH of ~60% (low $t_{LWIR}$). Compared with commercial white paint used in buildings and vehicles, our AMTA demonstrated obvious advantages in outdoor cooling.

The AMTA with a fairly low thickness presents a combination of high tensile strength and excellent optical properties, substantially superior to a variety of previously reported PDRC films (Fig. 5k)[40], offering a momentous candidate for the cooling requirements of outdoor devices. Besides, electrical insulation and thermal stability are common features of ANFs and Mica (Supplementary Figs. 23 and 26), allowing AMTA potentially helpful in the protection of high-temperature cables to avoid security problems like spontaneous combustion. To mimic real-world operating conditions, several simulated field tests were performed, such as thermal treatment, water rinsing, scratch treatment, and UV radiation (Supplementary Figs. 24, 25, and 27). The $\bar{R}_{solar}$ and tensile strength of AMTA could be maintained stably regardless of different harsh environments, revealing its great environmental durability. Utilizing the remarkable comprehensive performance, AMTA can be widely employed in various thermal management systems in our daily activities. For instance, the application scenarios of AMTA in car covers and mobile phone cases are demonstrated as shown in Fig. 5d–g. After running a 2 h experiment exposed to the sunlight, average temperature drop $\Delta T$ of 10 °C and 3 °C were observed for the covered devices, respectively. Such strong cooling effects established a safe and comfortable state for outdoor devices, leaving AMTA a promising PDRC material.

## Discussion

To summarize, we have proposed and demonstrated lamellar ANFs/Mica@TiO2 radiative coolers with excellent comprehensive performance via an effective "Solvent exchange-Reprotonation" processing strategy, towards real-world applications of PDRC. The two-step protonation transition not only promotes the transformation of nanofibers to strong microfibers but also shapes a dendritic network with fibrillar joints as cores, enabling overloaded scatterers to be stably grasped and orderly embedded into well-organized lamellar microstructure with superior mechanical strength of ~112 MPa and Young's modulus of ~4 GPa. Meanwhile, by simultaneously introducing core-shell Mica@TiO2 scatterers and interlamellar micropores into AMTA, strong multiple scattering at core-shell and shell-air interfaces yields a high $\bar{R}_{solar}$ of 92% at a quite low thickness of 25 μm. Further combined with a $\bar{\varepsilon}_{LWIR}$ of 87%, such a design can generate an average sub-ambient temperature drop of ~3.35 °C under direct sunlight. Besides, considering the harsh outdoor environment, long-term durability including high temperature, UV radiation, water rinsing and scratch damage have been integrated into AMTA. With respect to its overall performance combinations, AMTA possesses great superiority compared with commercial pearlescent coatings and currently existing polymeric PDRC materials, expecting it a prospective material for meeting long-term cooling requirements of outdoor devices.

## Methods
### Raw materials

Raw AMFs (Kevlar 49) were purchased from DuPont, USA. Mica, TiO2, and Mica@TiO2 were offered by Fujian Kuncai Material Technology Co. Ltd., China. Chemicals including DMSO, IPA, KOH, and APTES were all purchased from Aladdin Reagent Limited Corporation, China, and used without any further purifications.

### Fabrication of lamellar PDRC films

According to a previous report[41], 2 g raw AMFs and 2 g KOH were first added into 100 mL DMSO. After mechanically stirring for 7 days at room temperature, a uniform, viscous, and crimson ANFs dispersion was obtained and further diluted to 0.8 wt% by DMSO for subsequent use. As pretreatment, 4.5 g Mica@TiO2 was dispersed in the mixture of ethanol and deionized water (70 mL, 9:1 w/w), added 3 mL APTES, and refluxed at 70 °C for 4 h. After that, Mica@TiO2 dispersion was filtered and washed five times with deionized water. APTES-treated Mica@TiO2 was dispersed into DMSO (10 mg/mL) via ultrasonic 20 min and mixed with the obtained ANFs dispersion under mechanical stirring to form homogeneous flaxen ANFs/Mica@TiO2 sols with different mass ratios. Subsequently, the uniform sol was slowly dropped into the highly turbulent flow of IPA, which was formed by high-speed shearing of IPA in the IKA Magic Lab device (IKA Works Inc.) operating at 10,000 rpm (Solvent exchange)[42]. Based on the obtained stable dendritic colloid suspension, free-standing lamellar PDRC films were directly prepared via a layer-by-layer VAF method followed by soaking in a water bath for 24 h (Reprotonation). A series of lamellar PDRC films with different Mica@TiO2 loading had been fabricated, in which a 50 wt% sample was denoted as AMTA. For comparison, ANFs/Mica and ANFs/TiO2 composites were prepared by the similar method described above.

### Characterization

The digital pictures were photographed by iPhone SE (12MP Main) and the microscopic morphology was revealed with an optical microscope (OM, Olympus BX51, Japan), field emission scanning electron microscope (FESEM, Apero S HiVac, FEI, USA), transmission electron microscope (TEM, Tecnai G2 F20 S-TWIN) and atomic force microscope (AFM, Bruker, USA). Energy dispersive X-ray spectroscopy (EDS, Octane Elect Super, EDAX, USA) mapping was used to check the distribution of elements on the surface. The Zeta potential of Mica@TiO2 was characterized by Brookhaven Zeta PALS 190 Plus. X-ray photoelectron spectroscopy (XPS) spectra were characterized on an ESCA-Lab Xi+ (ThermoScientific, USA) using a monochromatic Al-Kα X-ray source. The component and structure of Mica@TiO2 as well as lamellar PDRC films were characterized by Fourier-transform infrared spectroscopy (FTIR, Nicolet 6700, ThermoScientific, USA) and X-ray diffraction (XRD, DY1291, Philips, Holland) with a Cu-Kα radiation source. Thermal stability of lamellar PDRC films was performed on a thermogravimetric analyzer (TGA, Q500, TA, USA) from 25 to 800 °C at a heating rate of 10 °C min⁻¹ under nitrogen conditions. The DC breakdown strength and volume resistivity of the prepared insulating PDRC films were carried out on a voltage-withstanding tester (DDJ-50 kV, Guance Electronics Co. Ltd., China) at room temperature with ~45% relative humidity and an ultra-high resistance micro-current tester (ZST-121, Zhonghang Times Instrument Co. Ltd., China), respectively. Thermal conductivity ($\lambda$) of lamellar PDRC films was calculated from $\lambda = \alpha \times \rho \times C_p$, where $\alpha$, $\rho$, and $C_p$, respectively correspond to thermal diffusivity, mass density, and specific heat. Laser flash analysis (LFA 467 Hyper Flash, Netzsch, Germany) was used to measure $\alpha$ as the voltage and pulse width were set to be 250 V and 200 μs, respectively.

### Mechanical testing

Tensile stress-strain curves were recorded using a universal test instrument (Instron 5967, USA) with a 500 N load cell at room temperature and relative humidity of ~50%. Based on the results of previous exploration (Supplementary Fig. 9) and relevant literature standards[34,43,44] (ASTM D882, etc.), appropriate testing conditions were selected to comprehensively and accurately reflect the mechanical properties of PDRC films. The sample length and width for all the PDRC films were 30 and 5 mm, respectively. The thickness of each tested sample strip was obtained by averaging thickness values at 3 to 5 different positions and at least five specimens were measured for all

different groups. The gauge distance and loading rate were 15 mm and 5 mm/min, respectively. For single-edge notched tensile tests (SENT)[36,45,46], the samples with the size of 30 mm × 10 mm × 25 μm were slightly notched to approximately one-third of their widths by slightly sliding a razor blade repeatedly, FESEM images (Supplementary Fig. 11b, c) show that the final notch radius is about 50 μm. Further tensile tests (Gauge distance -15 mm, loading rate -1 μm/s) were carried out to expand notches without fracture of the splines, while the crack propagation behaviors were observed with FESEM and EDS.

## Optical spectrum characterization

Optical characterization was conducted across UV to mid-infrared (MIR) wavelengths. The reflectivity ($R$) and transmissivity ($T$) were measured using a UV–VIS–NIR spectrometer (Lambda 1050, PerkinElmer, USA) in the range of 0.25–2.5 μm with an integrating sphere, while in the MIR range (2.5–25 μm) using a FTIR spectrometer (Nicolet iS50, ThermoScientific, USA) equipped with a diffuse gold integrating sphere (Pike Technologies, USA). The emissivity ($\varepsilon$) is equal to absorptivity, which is obtained from the equation $\varepsilon = 1 - R - T$.

The definition of average solar reflectivity ($\bar{R}_{\text{solar}}$) is given by:

$$\bar{R}_{\text{solar}} = \frac{\int_{0.3\,\mu m}^{2.5\,\mu m} I_{\text{solar}}(\lambda) \times R_{\text{solar}}(\lambda)\mathrm{d}\lambda}{\int_{0.3\,\mu m}^{2.5\,\mu m} I_{\text{solar}}(\lambda)\mathrm{d}\lambda} \tag{2}$$

where $\lambda$ is the wavelength, $I_{\text{solar}}(\lambda)$ is the reference direct normal spectral irradiance ASTM G173 under air-mass 1.5, representing the global solar intensity, and $R_{\text{solar}}(\lambda)$ is the spectral reflectivity of the surface.

Definition of average atmospheric window emissivity ($\bar{\varepsilon}_{\text{LWIR}}$) is given by:

$$\bar{\varepsilon}_{\text{LWIR}} = \frac{\int_{8\,\mu m}^{13\,\mu m} I_{\text{BB}}(T,\lambda) \times \varepsilon(T,\lambda)\mathrm{d}\lambda}{\int_{8\,\mu m}^{13\,\mu m} I_{\text{BB}}(T,\lambda)\mathrm{d}\lambda} \tag{3}$$

where $I_{\text{BB}}(T, \lambda)$ is the spectral irradiance emitted by a blackbody at temperature $T$ (here assumed to be 25 °C) shown in Fig. 4d, and $\varepsilon(T,\lambda)$ is the spectral thermal emissivity of the surface.

## Stability and durability tests

(1) To evaluate the thermal stability in long-term direct exposure to the sunlight, AMTA was sandwiched between two Al plates and placed on a heating stage at 180 °C for 8 h. (2) To mimic the unexpected rainstorm environment, AMTA was rinsed under the high-speed (-6 m/s) water jet for 8 h. (3) Scratch damage test was orthogonally applied onto the AMTA (8×8 cm) surface under a normal load, over a distance of 6 cm at a constant scratch rate of 2 cm/s. Scratch resistance tests[47] were performed in a cyclic reciprocating steel-wool wear apparatus, illustrated in Supplementary Fig. 25a. Rough steel-wool (Grade #0000, fiber width -25 μm) was tightly wrapped around the bottom of the scratch resistance tester (HSR-2M, Changzhou Ruipin Precision Instrument Co. Ltd., China) and fixed firmly by a steel wire. The clamping AMTA were reciprocally abraded under vertical forces of 5, 10, and 20 N for 1000 cycles, followed by the observation of surface morphology as well as the measurement of changes in strength and solar reflectivity, as shown in Supplementary Fig. 25c–g. (4) Accelerated UV aging tests were conducted in a homemade weathering chamber equipped with an iodine lamp (365 nm maximum intensity, UV irradiance of $10.0 \pm 0.5$ W/m$^2$) at 35 °C for 24, 48, 72, and 96 h, respectively. The UV radiation dosage of a 96 h test is equivalent to 1 year of Florida sunshine exposure (annual UV dosage of about 280 MJ/m$^2$), which is an international benchmark for UV durability tests. The $\bar{R}_{\text{solar}}$ and tensile strength of AMTA before and after treatment are shown in Supplementary Fig. 27e.

## Reporting summary

Further information on research design is available in the Nature Portfolio Reporting Summary linked to this article.

## Data availability

The authors declare that the data supporting the findings of this study are available within the article and Supplementary Information. Additional datasets related to this study are available from the corresponding authors upon request. Source data are provided in this paper.

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

## Acknowledgements

We gratefully acknowledge the National Key Research & Development Plan of China (2022YFA1205200, H.D.) for financial support.

## Author contributions

L.X. was associated with conceptualization, investigation, experiments, simulations, formal analysis, writing—original draft, review & editing, methodology, and resources. Y.W., C.C., and X.C. were associated with methodology and data curation. H.D. was associated with conceptualization, formal analysis, funding acquisition, writing—review & editing, resources, and project administration supervision. Q.F. was associated with formal analysis, writing—review & editing, resources, and project administration supervision.

## Competing interests

The authors declare no competing interests.
