## [Peer Review File · Nature Communications]

Thin lamellar films with enhanced mechanical properties for durable radiative coolingREVIEWER COMMENTS

Reviewer #1 (Remarks to the Author):

The paper reports a nacreous PDRC film that integrates ANFs network with core-shell TiO₂-coated mica microplatelet (Mica@TiO₂) scatterers. This film has been claimed to possess enhanced strength, durability and cooling capability without compromising optical performance. However, the reported results are more or less specific without solid evidence and explanation, furthermore, several issues appear and question the validity and significance of their claims.

1. The authors describe their composites as nacreous materials but fail to demonstrate the similar mechanical properties of components observed in natural nacre. One of the key features of biological nacre is its organic matrix, which is orders of magnitude more compliant than the tablets to achieve a near-uniform shear stress transfer. In contrast, the authors claim to have fabricated nacreous films using ultra-high strength ANFs as the organic matrix and Mica@TiO₂ as building block. The authors simply observed the internal morphology of the composites without carefully comparing the mechanical properties and dimensions of the components with those of biological nacre. Therefore, the composites cannot be called nacreous materials.

2. The mechanical tests used in the article do not follow experimental standards. Sample size, notch depth and loading speed are all important factors that can affect the results of a mechanical test, but the authors provide no explanation for how these parameters were selected. This lack of detail makes it difficult to conclude whether or not the reported results are reliable.

3. The authors claim that there is crack bridging and delamination in PDRC film, but provide no experimental evidence to support this assertion. These toughening mechanisms are critical to the natural nacre, and it cannot be assumed that PDRC films reproduce them simply by virtue of their brick-and-mortar microstructure.

4. As shown in Fig.2a, the stress of composite increases with the Mica@TiO₂ loading decrease from 90wt% to 10 wt%. While the composites without Mica@TiO₂ exhibit lower stress than those containing 10 wt% Mica@TiO₂. This change in trend is not discussed by the authors, leaving readers to speculate about what might be causing it.

Given these issues, it is difficult to recommend this work for publication.

Reviewer #2 (Remarks to the Author):

I read this work with great interest. The authors designed a nacreous PDRC film integrating aramid nanofibers (ANFs) network with core-shell TiO₂-coated mica microplatelet (Mica@TiO₂) scatterers following the "solvent exchange-reprotonation" processing method for durable radiative Cooling. The actual cooling capability performance of PDRC was also demonstrated in outdoor tests day and night times. The PDRC achieved an average sub-ambient temperature drop of ~3.35 °C under direct sunlight and an average temperature rise of ~6.11 °C for the night. I think the work is worthy in energy saving and thermal management applications.

However, the performance is still lower than the recently reported PRC. For example, phase change material integrated with radiative cooling revealed a sub-ambient temperature drop of ~6.3 °C for daytime (ACS Nano 2023, 17, 2, 1693). Similarly, the \bar{R}_{solar} of ~98%, which is higher than that of the present report (92%).

How stable are TiO₂ nanoparticles (NPs) with mica? Meanwhile, the uniform distribution of NPs needs valuable confirmation.

Reviewer #3 (Remarks to the Author):

The manuscript: "Thin Nacreous Films with Enhanced Mechanical Properties for Highly Efficient Durable Radiative Cooling" presents a new thin coating inspired by natural nacre for Radiative Cooling applications. This type of technology is highly relevant: Radiative Cooling essentially addresses global warming. The material presented here is based on mica and TiO₂, the fabrication process is interesting, and the final result resembles the microstructure of nacre. The requirements for radiative cooling coatings are stringent: High reflectivity, mechanical strength, heat stability

and resistance to UV from prolonged outdoor exposure. The manuscript presents results for all these properties and very interestingly, the proposed material offers unique combinations of mechanical strength, optical properties and robustness. I only have a few comments that the authors can probably address in a revised version.

- 1) The thickness of the material considered in this report is 25 micron. Is this thickness optimal for all properties? For example: one would think that thicker coatings would be stronger?
- 2) How does the scratch resistance compare with other types of optical coatings?
- 3) Is this coating assembled on a substrate? How is this material supposed to be used in applications? On thicker panels?
- 4) The section "Results" starts with a section that reads like an introduction. The results section should focus on results.
- 5) Fig. 2d: The authors show microcracks in the film, and they assume that these cracks appeared during the tensile test. The authors should show that it is indeed the case, and that the crack did not appear during the fabrication process.
- 6) Finally the manuscript contains typos, odd grammatical forms and bad choice of words. It should be carefully edited and proofread.

Point-by-Point Responses to Reviewers' Comments

We would like to appreciate above three reviewers for their constructive suggestions and insightful comments. Their suggestions help us improve the clarity and readability of the manuscript, as well as the rigorous and reliable aspects of experiments. Accordingly, we have revised the manuscript throughout by referencing literature standards, conducting further experiments and discussions to address the queries of the reviewers. The point-by-point responses to the reviewers' comments are detailed below (blue color for responded texts and the red color for revised texts).

Reviewer #1 (Remarks to the Author):

The paper reports a nacreous PDRC film that integrates ANFs network with core-shell TiO₂-coated mica microplatelet (Mica@TiO₂) scatterers. This film has been claimed to possess enhanced strength, durability and cooling capability without compromising optical performance. However, the reported results are more or less specific without solid evidence and explanation, furthermore, several issues appear and question the validity and significance of their claims.

1. The authors describe their composites as nacreous materials but fail to demonstrate the similar mechanical properties of components observed in natural nacre. One of the key features of biological nacre is its organic matrix, which is orders of magnitude more compliant than the tablets to achieve a near-uniform shear stress transfer. In contrast, the authors claim to have fabricated nacreous films using ultra-high strength ANFs as the organic matrix and Mica@TiO₂ as building block. The authors simply observed the internal morphology of the composites without carefully comparing the mechanical properties and dimensions of the components with those of biological nacre. Therefore, the composites cannot be called nacreous materials.

Response: We sincerely appreciate the careful reading of the manuscript. As the reviewer suggested, the organic matrix, which assumes an important role in natural nacre for a near-uniform shear stress transfer, is typically smaller in size than the tablets and several orders of magnitude less compliant (*Nat. Mater.* **2015**, *14*, 23-26). Given that we have employed high-modulus assemblies for both organic (ANFs ~5-10 GPa; *Adv. Funct. Mater.* **2020**, *30*, 2000186) and inorganic (Mica@TiO₂ ~178 GPa; *J. Phys.: Condens. Matter* **1993**, *5*, 1681; *Nano Res.* **2012**, *5*, 550-557) components to deal with harsh outdoor environments, our previous designation of “nacreous materials” to refer

to the newly developed PDRC films reported in the manuscript is indeed lack of careful consideration, since there is no significant order of magnitude difference in the strength, flexibility and dimension of these components.

Therefore, we replaced the previous designation “nacreous structure” with a more realistic “lamellar structure with interlayer micropores”, which is also in line with our original intention of enhancing mechanical properties as well as light scattering. Also, to more clearly address the reviewer’s question, here we would like to **highlight the design and novelty of this work from the following three aspects:**

(1) “Solvent exchange-Reprotonation” processing strategy.

Aramid nanofibers (ANFs), as the deprotonation products of aramid fibers, are more conducive to processing, but it is difficult to reproduce the theoretical ultrahigh strength (*Adv. Funct. Mater.* **2020**, *30*, 2000186; *Composites Part A* **1998**, *29*, 1411-1415) of aramid fibers in secondary processing, i.e., the precise regulation of the reprotonation process of ANFs remains a problem (*ACS Nano* **2022**, *16*, 5984-5993; *Adv. Mater.* **2020**, *32*, 1906939; *ACS Nano* **2015**, *9*, 2489-2501).

Herein, we demonstrate a novel “Solvent exchange-Reprotonation” processing strategy to have more control on the reprotonation path and morphological evolution of ANFs. Specifically, during solvent exchange and turbulent shear in IPA, the rather weak protonation environments and strong fluid shear forces contributed to the formation of the dendritic ANFs network with fibrillar joints (Fig. 1d,e and Supplementary Fig. 6), where overloaded Mica@TiO₂ (> 50 wt%) scatterers can be stably grasped and loaded for several hours (Supplementary Fig. 7). Subsequent vacuum-assisted deposition resulted in a well-organized lamellar structure in which the 2D Mica@TiO₂ microplatelets were anchored in alignment (Supplementary Fig. 1h). Eventually, through the regeneration effect of strong proton donor (water), a 3D shear stress transfer network of dendritic microfibers was successfully constructed, resulting in mechanical properties superior to those of conventional processing methods (Fig. 2a). Moreover, the formation of well-organized lamellar structures also relies on the ordered-stacking of finely 2D sheet-like scatterers with the help of dendritic ANFs networks, which tend to exhibit higher mechanical strength compared to 0D scatterers. Please see Supplementary Fig. 15a, where we have also added a detailed description in the figure caption (red texts) for greater clarity.

Fig. 1d,e and h:

Fig. 1 | Illustration of the fabrication process and microstructure characterizations. d,e OM (d) and FESEM (e) images of ANFs network with fibrillar joints, showing hyperbranched morphology. **h** Section-view FESEM image of AMTA, showing a well-organized lamellar structure with interlayer micropores.

Supplementary Fig. 7:

Supplementary Fig. 7. Stability of the ANFs/Mica@TiO₂ suspensions. Front and bottom photographs were taken at different moments after the suspensions were prepared. From left to right, four samples in photographs are respectively (1) ANFs/DMSO/Mica@TiO₂/IPA processed by “Solvent exchange-Turbulent shear”, (2) ANFs/DMSO/Mica@TiO₂/H₂O processed by “Solvent exchange-Stirring” (conventional methods), (3) ANFs/DMSO/Mica@TiO₂ without solvent exchange and (4) Mica@TiO₂/DMSO. All suspensions contain the same Mica@TiO₂ loading, i.e. 90 wt%. It can be observed that the suspension processed by our method could be stably dispersed for several hours without any filler sedimentation, indicating the effect of hyperbranched ANFs network with fibrillar joints on trapping Mica@TiO₂ within the suspension. Furthermore, although the ANFs network would also deposit with Mica@TiO₂ to a certain extent after a long time, but it could easily return to the original state only by gently turning it over, providing an industrial prospect for the subsequent preparation of film or slurry.

Fig. 2a:

Fig. 2 | Mechanical properties and multiscale extrinsic toughening mechanisms. a Tensile stress-strain curves of ANFs/Mica@TiO₂ composites via different preparation methods.

Supplementary Fig. 15a:

Supplementary Fig. 15. Effect of different dielectric scatterers on mechanical properties.

a) Tensile stress-strain curves of PDRC films with different dielectric scatterers. For different types of dielectric scatterers, such as Mica@TiO₂, Mica and TiO₂, their intrinsic geometry has significant impact on the mechanical performance of these PDRC films. Generally speaking, 2D platelet scatterers with inherent ultra-high aspect-ratio, like Mica@TiO₂ and Mica, exhibit outstanding mechanical properties, especially in strength and modulus. At the same time, the well-organized lamellar microstructure assembled with 2D platelets and ANFs network can further strengthen and toughen these PDRC films. However, for TiO₂ scatterers, the agglomeration phenomenon caused by high specific surface energy of zero-dimensional nanoparticles leads to poor mechanical performance and weak light scattering effect.

(2) Multiple scattering at interfaces of core-shell Mica@TiO₂ and interlamellar micropores.

In the field of PDRC, the desired high solar reflectivity mainly depends on the extent of multiple sunlight scattering, which is caused by the refractive index contrast at the interface between two phases, that is, impedance mismatch (*Adv. Funct. Mater.* **2022**, *32*, 2109542; *Electronics* **2019**, *8*, 1022).

Herein, we innovatively combine two design principles, i.e., the simultaneous introduction of both dielectric scatterers and hierarchical porous structures, so as to improve the scattering intensity and efficiency in a limited space, and ultimately achieve higher solar reflectivity without compromising other applicable performance. Specifically, the high refractive index Mica@TiO₂ core-shell scatterer has a more desirable backscattering behavior than Mica or TiO₂ alone by virtue of its intrinsically existing scattering interfaces, especially at ~500 nm where the solar irradiation is the strongest (Fig. 3 and Supplementary Figs. 17 and 18). Similar to the literature reports (*Science* **2018**, 362, 315-319; *Nano Energy* **2021**, 81, 105600; *Nat. Commun.* **2021**, 12, 365), the appropriately sized lamellar micropores (~5 μm) induced by solvent evaporation also contribute to the superior optical properties through lamellar interface scattering (Supplementary Figs. 19 and 20). All the above conclusions were verified by optical simulations and relevant experiments. Therefore, the strong multiple scattering behaviors at core-shell and shell-air interfaces within AMTA yield a high solar reflectivity of 92% at a quite low thickness of 25 μm.

Fig. 3b-d:

Fig. 3 | Optical properties comparison among Mica@TiO₂, Mica and TiO₂. b-c Simulated near-field electric field distributions at 0.5 μm wavelength (**b**) and far-field scattering phase function at 0.5 μm wavelength (**c**) for different monodisperse scatterers, where the size consistent with real situation. E and k represent electric field and wave vector of the incident light, respectively. 90° denotes as incident direction. **d** Schematic illustration of multiple scattering behaviors of AMTA caused by dielectric contrast, including Mica@TiO₂ and interlaminar micropores.

Supplementary Fig. 20a:

Supplementary Fig. 20. Verification of the interlaminar micropores on solar reflectivity for AMTA by FEM simulations. a) Simulated section-view electric field norm distributions, scale bar, 3 μm . More significant scattering behaviors are present in the “Large pore” structure ($\sim 5 \mu\text{m}$) and tend to be backward, in agreement with the trend of the experimentally measured optical reflectivity (Supplementary Fig. 19).

(3) Comprehensive performance for real-world applications of PDRC.

When it comes to the cooling requirements of outdoor devices, it is far from enough to meet the basic optical performance. In particular, PDRC materials that have been serviced in hot outdoors for a long time are bound to face challenges such as mechanical damage, UV weathering and high-temperature heating (*EcoMat* **2022**, 4, e12153; *Adv. Funct. Mater.* **2022**, 32, 2206962). Therefore, how to achieve superior mechanical strength and environmental durability without compromising the optical properties, i.e., a balance, remains a problem to be solved (*Nat. Commun.* **2022**, 13, 4805; *Adv. Mater.* **2022**, 34, 2208236).

Herein, we have successfully prepared a new class of PDRC films/coatings by combining the ideas mentioned in the previous two parts. It not only reaches the basic optical solar reflectivity required by PDRC, but also has a great advantage in terms of mechanical strength, which is apparently different from other reports (*Adv. Funct. Mater.* **2022**, 32, 2109542; *Adv. Sci.* **2022**, 9, 2202061; *Proc. Natl. Acad. Sci. U.S.A.* **2020**, 117, 205-213; *Adv. Mater. Technol.* **2022**, 7, 2100803). Moreover, relying on the intrinsic superiority of the internal components (e.g., high hardness and high temperature resistance of ANFs and Mica, UV shielding of TiO_2 , etc.), AMTA also exhibited excellent environmental durability, including UV radiation, high temperature and scratches (Supplementary Figs. 24-27), which is of great significance for the practical application of PDRC materials.

2. The mechanical tests used in the article do not follow experimental standards. Sample size, notch depth and loading speed are all important factors that can affect the results of a mechanical test, but the authors provide no explanation for how these parameters were selected. This lack of detail makes it difficult to conclude whether or not the reported results are reliable.

Response: Thank you for the valuable suggestion. According to the reviewer's comments, we further consulted the relevant literature (e.g., *Nat. Mater.* **2022**, *21*, 1121-1129; *Adv. Mater.* **2023**, *35*, 2300241; *Science* **2021**, *374*, 96-99) and industrial standards (ASTM D882, etc.) for mechanical testing of thin films, and conducted additional experiments and detailed discussions to ensure that our results are accurate and reliable. As mentioned in the suggestions, sample size and loading rate are all important factors that can affect the results of a mechanical test. In order to determine the optimal experimental conditions, we first investigated the influence of different test factors on mechanical performance, specifically sample width, gauge distance, sample thickness and loading rate, and each factor was assigned three values within the range commonly used in literature and standards, as shown in Table R1.

Table R1 | Mechanical testing conditions.

Test No.	Sample width (mm)	Gauge distance (mm)	Sample thickness (μm)	Loading rate (mm/min)
1	3	15	25	5
2	5	15	25	5
3	10	15	25	5
4	5	5	25	5
5	5	10	25	5
6	5	15	25	5
7	5	15	10	5
8	5	15	25	5
9	5	15	50	5
10	5	15	25	1
11	5	15	25	5
12	5	15	25	10

For the stress-strain curves under different test conditions, please see Supplementary Fig. 9.

Supplementary Fig. 9:

Supplementary Fig. 9. Tensile stress-strain curves of AMTA under different mechanical testing conditions. a-d) Sample width (3, 5, 10 mm). e-h) Gauge distance (5, 10, 15 mm). i-l) Sample thickness (10, 25, 50 μm). m-p) Loading rate (1, 5, 10 mm/min). Based on the above test results and literature standards (ASTM D882, etc.), sample width ~ 5 mm, gauge distance ~ 15 mm, sample thickness $\sim 25 \mu\text{m}$ and loading rate ~ 5 mm/min were selected as the final testing conditions, which can reflect the mechanical properties of AMTA more accurately and comprehensively.

As shown in the above test results, the loading rate and sample thickness have a strong influence on the final mechanical properties, while the effects of sample width and gauge distance are relatively small. So we assigned one of the most representative test conditions for the subsequent mechanical tests, the exact implementation of which

was detailed in the “Mechanical testing” section on the manuscript page 14-15 (in addition, please see comment 3 for single-edge notched tensile tests and corresponding crack propagation behaviors).

Page 14-15, “Mechanical testing”:

“Tensile stress-strain curves were recorded using a universal test instrument (Instron 5967, USA) with a 500 N load cell at room temperature and relative humidity of ~50%. Based on the results of previous exploration (Supplementary Fig. 9) and relevant literature standards (ASTM D882, etc.), appropriate testing conditions were selected to comprehensively and accurately reflect the mechanical properties of PDRC films. The sample length and width for all the PDRC films were 30 and 5 mm, respectively. The thickness of each tested sample strip was obtained by averaging thickness values at 3 to 5 different positions and at least five specimens were measured for all different groups. The gauge distance and loading rate were 15 mm and 5 mm/min, respectively.”

Furthermore, we also collated several raw stress-strain curves for each set of samples in Supplementary Fig. 10, which demonstrated the excellent repeatability and reliability of our reported PDRC materials.

Supplementary Fig. 10:

Supplementary Fig. 10. Tensile stress-strain curves of ANFs/Mica@TiO₂ composites. a-f) Tensile stress-strain curves of ANFs/Mica@TiO₂ composites with different scatterer loading (0, 10, 30, 50, 70, 90 wt%). **g,h)** Photographs of AMTA before (**g**) and after (**h**) stretching under

uniaxial tensile testing.

3. The authors claim that there is crack bridging and delamination in PDRC film, but provide no experimental evidence to support this assertion. These toughening mechanisms are critical to the natural nacre, and it cannot be assumed that PDRC films reproduce them simply by virtue of their brick-and-mortar microstructure.

Response: Thank you for the kind suggestion. Understanding the failure mechanism of materials during stretching is critical for the design and application of high mechanical performance PDRC films. To demonstrate the positive effect of the well-organized lamellar microstructure within AMTA on its mechanical properties, including toughening mechanisms, the single-edge notched tensile tests (SENT, for the observation of crack propagation behaviors) and corresponding finite-element simulations were performed, as shown in Fig. 2d, Supplementary Figs. 11 and 12. Please see the “Mechanical testing” section on the manuscript page 14-15 for the exact implementation of SENT and the “Note S4: Microscale Finite-Element Analysis of Mechanical Response” section on the supplementary materials page 5 for the modelling principles.

Page 14-15, “Mechanical testing (SENT)”:

“For single-edge notched tensile tests (SENT, *Nat. Commun.* **2017**, *8*, 287; *Matter* **2019**, *1*, 412-427; *Nat. Commun.* **2021**, *12*, 4539), the samples with the size of 30 mm × 10 mm × 25 μm were slightly notched to approximately one-third of their widths by slightly sliding a razor blade repeatedly, FESEM images (Supplementary Fig. 11b,c) show that the final notch radius is about 50 μm. Further tensile tests (Gauge distance ~15 mm, loading rate ~1 μm/s) were carried out to expand notches without fracture of the splines, while the crack propagation behaviors were observed by FESEM and EDS.”

During stretching, the crack in AMTA initiates from the notch and propagates along a tortuous path (Fig. 2d and Supplementary Fig. 11b,c). First of all, failures in the crack propagation path, such as microscopic deformation and curling, can be definitely observed both within (Supplementary Fig. 11d-f) and between (Fig. 2d₁, Supplementary Figs. 11g and 13) the layers. The in-plane layers exhibit strong plastic deformation of dendritic ANFs network (Supplementary Fig. 11d-f), while between the layers, a trapezoid-like fracture cross-section (Fig. 2d₁, Supplementary Figs. 11g and 13) is formed with joint participation of the Mica@TiO₂ and microfiber networks. Both phenomena demonstrate a complex multi-stage ductile fracture process within AMTA,

rather than a simple one-time brittle fracture, which is essential for materials in long-term outdoor service. Meanwhile, the soft (ANFs ~5-10 GPa) and rigid (Mica@TiO₂ ~178 GPa) two-component network allows the crack to deflect (Fig. 2d₂ and Supplementary Fig. 11h,j) and branch (Fig. 2d₃ and Supplementary Fig. 11e,j) whenever the embedded Mica@TiO₂ microplatelets are encountered, resulting in abundant tiny cracks uniformly distributed inside adjacent layers (Fig. 2d₁, Supplementary Figs. 11g and 13). Although deflection in AMTA is not as significant as that produced by the “brick-and-mortar” structure in natural nacre (*Nat. Mater.* **2015**, *14*, 23-26), it does play a role in dispersing and transferring stress during stretching, i.e., as an important part of the failure analysis mechanism. With strain further increases, Mica@TiO₂ that spans the whole major crack to withstand more stress (*Adv. Mater.* **2023**, *35*, 2300241; *ACS Nano* **2020**, *14*, 611-619) and branched cracks that bridged at the tip for extrinsic toughening have also been observed (Supplementary Fig. 11h-j).

Moreover, nonlinear finite element modeling simulations (Supplementary Fig. 12) also show the mechanical response of AMTA during uniaxial stretching. In particular, both the ANFs (cohesive zones) and micropores undergo noticeable tensile deformation (Supplementary Fig. 12c,d) to dissipate stresses, while crack deflection and interlamellar slip are correspondingly observed in the 2D cross-section of the notched model (Supplementary Fig. 12d), which resulted from the well-organized lamellar structure stacked by the ANFs network after gripping Mica@TiO₂ microplatelets. These simulated structural features under crack propagation are consistent with those we observed in AMTA.

Fig. 2d:

Fig. 2 | Mechanical properties and multiscale extrinsic toughening mechanisms. d Crack

propagation behaviors of AMTA observed from FESEM image and EDS elemental maps of C, Si and Ti, showing long-range crack deflection and branching, as well as strong plastic deformation and curling of dendritic ANFs adhesives.

Supplementary Fig. 11:

Supplementary Fig. 11. Crack propagation behaviors of AMTA in single-edge notched tensile (SENT) tests. **a)** Schematic illustration of SENT experiments. **b,c)** FESEM images of notched AMTA before **(b)** and after **(c)** stretching, demonstrating that circuitous crack propagation occurred indeed after stretching the notch. **d)** major crack next to notch. **e)** Pull-out and fracture of Mica@TiO₂ microplatelets. **f,g)** Plastic deformation and curling of dendritic ANFs adhesives at the edge of cracks. **h,i)** Mica@TiO₂ microplatelets span the whole major crack to withstand more stress. **j)** Tip of the long microcrack, showing short cracks were bridged by Mica@TiO₂ on both sides.

Supplementary Fig. 12:

Supplementary Fig. 12. Microscale Finite-Element Analysis of Mechanical Response of AMTA. **a)** Bilinear traction-separation response of the CZM. **b)** Tensile stress-strain curves of AMTA with porous lamellar microstructures under uniaxial tension mode from experiments and FEM simulation. **c,d)** Stress nephograms of AMTA with **(d)** and without **(c)** a single edge notch during structural deformation and failure process under uniaxial tension. Specifically, significant tensile deformation of cohesive zones (ANFs) and pores occurred, dissipating a large amount of stress, while external toughening by microcracks deflection and bridging was also observed in the Mica@TiO₂-stacked lamellar microstructures.

After the complete failure of AMTA, the fracture morphology also reveals the strong plastic deformation and curling of dendritic microfiber networks (Supplementary Fig. 13c,d) and the pull-out of Mica@TiO₂ microplatelets (Supplementary Fig. 13a,b), which act as efficient energy dissipation modes by frictional sliding and breaking of polymer networks.

All in all, through the detailed discussion of the above experimental results and mechanical simulations, the internal toughening mechanisms of AMTA have been better clarified, which share common features with the toughening effects of natural nacre. But here, it is more the dendritic microfiber (ANFs) networks within the lamellar microstructure that play a key role, which is different from other highly tough materials.

Supplementary Fig. 13:

Supplementary Fig. 13. Inclined-view FESEM images of the tensile fracture surface of AMTA. a) Overview of the fracture surface, showing lamellar trapezoidal structure. **b)** Pull-out of Mica@TiO₂ microplatelets and extensive interface delamination toughening. **c,d)** Strong plastic deformation and curling of dendritic ANFs adhesives.

4. As shown in Fig.2a, the stress of composite increases with the Mica@TiO₂ loading decrease from 90wt% to 10 wt%. While the composites without Mica@TiO₂ exhibit lower stress than those containing 10 wt% Mica@TiO₂. This change in trend is not discussed by the authors, leaving readers to speculate about what might be causing it.

Response: Thank you for the kind suggestion. As noted by the reviewer, with increasing scatterer content, the mechanical strength of PDRC films illustrate a tendency to increase first and the decrease, reaching a maximum at about 10 wt%. This is mainly because the introduction of a small amount of Mica@TiO₂ microplatelets contributes to the formation of a well-organized lamellar structure, which can be further strengthened by the toughening mechanisms mentioned in comment 3, resulting in better mechanical performance. However, with the incorporation of an excessive amount of large-size scatterers (diameter ~20 μm), micropores are inevitably introduced into lamellar structures. Although a suitable combination of porosity and pore-size (~5 μm, e.g., *Science* **2018**, 362, 315-319; *Nano Energy* **2021**, 81, 105600; *Nat. Commun.* **2021**, 12, 365) can significantly enhance the light scattering behaviors (Supplementary Figs. 19 and 20), overly increased pore-size (Fig. R1) obviously leads more defects, thus weakens the mechanical properties to a great extent. The above discussion has been revised on manuscript page 5.

Fig. R1:

Fig. R1. Section-view FESEM image of PDRC films with 90 wt% Mica@TiO₂, showing an oversized (~10 μm) microporous structure with scattered distribution of Mica@TiO₂ microplatelets.

Reviewer #2 (Remarks to the Author):

I read this work with great interest. The authors designed a nacreous PDRC film integrating aramid nanofibers (ANFs) network with core-shell TiO₂-coated mica microplatelet (Mica@TiO₂) scatterers following the “solvent exchange-reprotonation” processing method for durable radiative Cooling. The actual cooling capability performance of PDRC was also demonstrated in outdoor tests day and night times. The PDRC achieved an average sub-ambient temperature drop of ~3.35 °C under direct sunlight and an average temperature rise of ~6.11 °C for the night. I think the work is worthy in energy saving and thermal management applications.

1. However, the performance is still lower than the recently reported PRC. For example, phase change material integrated with radiative cooling revealed a sub-ambient temperature drop of ~6.3 °C for daytime (ACS Nano 2023, 17, 2, 1693). Similarly, the \bar{R}_{solar} of ~98%, which is higher than that of the present report (92%).

Response: Thank the reviewer very much for his or her positive and encouraging comments, and we are equally inspired by these kind suggestions. We have carefully and thoroughly studied this recently reported article (ACS Nano 2023, 17, 1693-1700) on temperature-adaptive thermal management. The authors have combined PDRC materials and room-temperature phase change materials to construct a novel all-weather thermal management device with outstanding performance in daytime cooling (-6.3 °C) and nighttime warming (+2.1 °C). Especially in the aspect of the solar reflectivity mentioned by the reviewer, the porous expanded polytetrafluoroethylene (ePTFE,

~98%) used in above research work does have superior optical performance compared to the AMTA (~92%) we reported here. However, as we discussed in the “**Introduction**”, while the optical properties of PDRC materials are important, the comprehensive performance during their service life, including mechanical properties, environmental durability, etc., must also be taken into consideration at the same time, as it is crucial for the future practical application of PDRC. Therefore, we believe that the present report is taking a different perspective on the design and application of PDRC materials. Although the current research still has some minor flaws, it certainly provides a new idea for the real-world applications of PDRC. In addition, we are also conducting some related investigations in parallel to address the issues mentioned by the reviewer, and hope to make a greater contribution to the field in the near future. Related discussion and literature have been added into the manuscript.

2. How stable are TiO₂ nanoparticles (NPs) with mica? Meanwhile, the uniform distribution of NPs needs valuable confirmation.

Response: Thank you for the kind suggestion. The coating stability of TiO₂ nanograins on Mica surface have a significant impact on the light scattering behaviors of Mica@TiO₂ at the refractive index contrast interfaces. Therefore, we performed additional experiments to examine the combination stability of both Mica and TiO₂, including XPS analyses and vigorous ultrasonic tests, as shown in Supplementary Fig. 4. The XPS full spectra of Mica, TiO₂ and Mica@TiO₂ (Supplementary Fig. 4a) show that the Mica@TiO₂ core-shell scatterer has the characteristic elemental peaks (Ti2p and Si2p) of both Mica and TiO₂, which proves the successful combination of them. Further analysis of O1s, Ti2p and Si2p fine spectra of the three can obtain detailed chemical bonding information at the contact interface between Mica and TiO₂. Specifically, In the O1s spectrum of Mica@TiO₂, a new Ti-O-Si peak (531.1 eV) can be observed, while the peak shift of Ti2p_{3/2} (+0.30 eV) and Si-O-Si (-0.45 eV) is also captured in the Ti2p and Si2p spectra, respectively, which indicates that the Mica and its surface TiO₂ nanograins are not only physical binding, but a new type of chemical bonding (Ti-O-Si) is formed at the interface fusion, so that the two stable combination. For the vigorous ultrasonic tests (10 h, 600 W), we can also distinctly observe that the surface morphology and elemental distribution (Si and Ti) of Mica@TiO₂ did not undergo significant changes before and after ultrasonic through FESEM and corresponding EDS images (Supplementary Fig. 4b,d). In particular, the TiO₂

nanograins on the Mica surface also did not produce any loosening or shedding (Supplementary Fig. 4c,e), proving the outstanding bonding stability of both.

Supplementary Fig. 4:

Supplementary Fig. 4. Stability of the combination of Mica and TiO₂. a) XPS full spectra and O1s, Ti2p, Si2p fine spectra of Mica, TiO₂ and Mica@TiO₂. In the O1s spectrum of Mica@TiO₂, a new Ti-O-Si peak (531.1 eV) can be observed, while the peak shift of Ti2p_{3/2} (+0.30 eV) and Si-O-Si (-0.45 eV) is also captured in the Ti2p and Si2p spectra, respectively, which indicates that the Mica and its surface TiO₂ nanograins are not only physical binding, but a new type of chemical bonding (Ti-O-Si) is formed at the interface fusion, so that the two stable combination. b-e) FESEM images of Mica@TiO₂ (b,d) and TiO₂ nanograins (c,e) before and after long-term (10 h, 600 W) ultrasonic testing, showing that the TiO₂ on Mica surface did not undergo significant loosening or shedding under high-power ultrasonic damage, demonstrating the excellent bonding stability of both.

Meanwhile, after the initial observation of TiO₂ nanograins distributed on Mica surface by FESEM images, we further carefully characterized the microstructure of the Mica@TiO₂ core-shell scatterer by TEM, as shown in Supplementary Fig. 3. The Mica surface was completely coated by TiO₂ nanograins, forming a dense cladding layer

(Supplementary Fig. 3a). With further magnification, the uniform coating produced by inter-bonding of granular TiO₂ can be apparently observed at the edges and intermediate regions (Supplementary Fig. 3b,c), directly demonstrating their uniform and dense distribution, while high-magnification images at the Mica edges (Supplementary Fig. 3d) also prove that the thickness of the TiO₂ layer is about less than ~100 nm. In addition, HRTEM (Supplementary Fig. 3e,f) and corresponding fast Fourier transform (inset images) of TiO₂ nanograins clearly indicate that the two types of diffraction patterns and lattice fringes are consistent with the (110) and (200) planes of the Rutile TiO₂ (*Phys. Chem. Chem. Phys.* **2013**, *15*, 10978-10988), which is an ideal inorganic scatterer with higher refractive index ($k \sim 2.7$), laying the foundation for multiple scattering behaviors and high solar reflectivity of AMTA.

Supplementary Fig. 3:

Supplementary Fig. 3. Characterization of the distribution and crystalline phase of TiO₂ on Mica surface. a) TEM image of the Mica@TiO₂, indicating that the mica platelet is completely coated with TiO₂ nanograins, which in turn forms a multi-interfaces core-shell structure. b-d) Corresponding magnified TEM images of TiO₂ nanograins on Mica surface, directly demonstrating their uniform and dense distribution, while high-magnification images at the Mica edges also prove that the thickness of the TiO₂ layer is about less than ~100 nm. e,f) High-resolution TEM images (HRTEM) and corresponding fast Fourier transform (inset images) of TiO₂ nanograins. Two types of diffraction patterns and lattice fringes are consistent with the (110) and (200) planes of the Rutile TiO₂, which is an ideal inorganic scatterer with higher

refractive index ($k \sim 2.7$).

Reviewer #3 (Remarks to the Author):

The manuscript: “Thin Nacreous Films with Enhanced Mechanical Properties for Highly Efficient Durable Radiative Cooling” presents a new thin coating inspired by natural nacre for Radiative Cooling applications. This type of technology is highly relevant: Radiative Cooling essentially addresses global warming. The material presented here is based on mica and TiO_2 , the fabrication process is interesting, and the final result resembles the microstructure of nacre. The requirements for radiative cooling coatings are stringent: High reflectivity, mechanical strength, heat stability and resistance to UV from prolonged outdoor exposure. The manuscript presents results for all these properties and very interestingly, the proposed material offers unique combinations of mechanical strength, optical properties and robustness. I only have a few comments that the authors can probably address in a revised version.

1. The thickness of the material considered in this report is 25 micron. Is this thickness optimal for all properties? For example: one would think that thicker coatings would be stronger?

Response: Thank the reviewer very much for his or her positive comments and valuable suggestions. Regarding the concerns of the reviewer, thickness is always an important issue in the field of coatings or films that cannot be avoided, as it relates to the cost, comprehensive performance, aesthetics and convenience of the material. It is easy to assume that thicker materials are indeed more difficult to break, i.e., “stronger”, both in terms of apparent external forces and environmental weathering. Even in optical performance, greater thicknesses give relatively better solar reflectivity and infrared emissivity (but subject to an upper limit). For example, we explored the influence of thickness on the mechanical properties and optical reflectivity of AMTA, as shown in Supplementary Fig. 9i-1 and Fig. 4a. For mechanical strength, thickness has little effect (but the apparent external force is positively correlated with thickness), while the solar reflectivity increases with thickness (higher transmissivity at lower thickness) and eventually level off (> 25 micron).

Supplementary Fig. 9i-1:

Supplementary Fig. 9. Tensile stress-strain curves of AMTA under different mechanical testing conditions. i-l) Sample thickness (10, 25, 50 μm).

Fig. 4a:

Fig. 4 | Comprehensive optical properties and theoretical cooling performance. a Thickness optimization of AMTA based on the solar reflectivity.

However, for materials oriented to practical applications, or products, indicators such as cost, aesthetics and portability (lightweight) are crucial, especially for products like PDRC devices against outdoor service, where their importance sometimes exceeds the pivotal strength and optical properties (*Joule* **2020**, *4*, 1350-1356; *Nano Lett.* **2022**, *22*, 4925-4932; *Adv. Funct. Mater.* **2022**, *32*, 2206962). Because intrinsic properties (strength, optical properties, etc.) only need to meet the basic indicators of the target scenario, in fact, does not necessarily reach particularly high. Lower thickness, indeed, often means less cost, better appearance and comfort (lightweight), as well as a wider range of application scenarios, making the ultimate industrialization of PDRC materials much more probable. Therefore, minimizing the thickness of the material while meeting the mandatory specifications (e.g., strength, hardness, optical performance, weathering resistance, etc.), i.e., the balance of the above properties, is of great significance for PDRC coatings/films.

As for our present report, the 25 micron AMTA integrates 112 MPa strength, 4 GPa Young's modulus, 92% solar reflectivity and corresponding various environmental resistance. This balance of comprehensive performance is dramatically different from

other reports, providing positive significance for the practical applications of PDRC.

2. How does the scratch resistance compare with other types of optical coatings?

Response: Thank you for the kind suggestion. We conducted scratch resistance tests on the AMTA coating with reference to the methods and standards in the literature (*Adv. Mater.* **2017**, *29*, 1700205; *ACS Appl. Mater. Interfaces* **2022**, *14*, 39432-39440), please see the “Stability and durability tests” section on manuscript page 16 for specific implementation, and the test results are shown in Supplementary Fig. 25.

Page 16, “Stability and durability tests (scratch resistance)”:

“Scratch resistance tests were performed in a cyclic reciprocating steel wool wear apparatus, illustrated in Supplementary Fig. 25a. Rough steel-wool (Grade #0000, fiber width $\sim 25\ \mu\text{m}$) was tightly wrapped around the bottom of the scratch resistance tester (HSR-2M, Changzhou Ruipin Precision Instrument Co. Ltd., China) and fixed firmly by a steel wire. The clamping AMTA were reciprocally abraded under vertical forces of 5, 10 and 20 N for 1000 cycles, followed by the observation of surface morphology as well as the measurement of changes in strength and solar reflectivity, as shown in Supplementary Fig. 25c-g.”

Supplementary Fig. 25:

Supplementary Fig. 25. Scratch resistance of AMTA. **a)** Front and top views of cyclic scratch experiments with a steel wool. **b)** The friction coefficient (~ 0.25) of AMTA under scratch load of 10 N for 1000 cycles, which is comparable to that of glass and lower than that of most common polymers, demonstrating the excellent scratch resistance of AMTA. **c)** The almost unchanged \bar{R}_{solar} and tensile strength of AMTA after bearing 0, 5, 10 and 20 N scratch load for 1000 cycles. **d-g)** FESEM images of AMTA after severe scratch testing, exhibiting no fatal marks on surface even under 10 N scratch load.

Editorial Note: Fig. R2 below reproduced with permission from Li et al., Thermo-optically designed scalable photonic films with high thermal conductivity for subambient and above-ambient radiative cooling. *Adv. Funct. Mater.* **32**, 2109542 (2022). Copyright John Wiley & Sons.

Under 10 N scratch load, due to the presence of high hardness aramid and Mica, the surface of AMTA did not produce any prominent damage, and exhibited a fairly low friction coefficient of 0.25 (Supplementary Fig. 25b,f), which is lower than that of most common polymers (> 0.3 , *Wear* **2008**, 265, 606-611; *Mater. Lett.* **2005**, 59, 175-179; *Wear* **1996**, 200, 137-147; *Wear* **1966**, 9, 329-348), providing the excellent scratch resistance of AMTA. Additionally, further testing revealed that the solar reflectivity and tensile strength before and after scratching remained essentially stable (Supplementary Fig. 25c). Only at 20 N scratch load, the strength of AMTA resulted a certain weakening, which mainly stemmed from the wrinkling damage (Supplementary Fig. 25g) of its surface ANFs network under vertical pressure.

Since many optical coatings reported in the literature (*Adv. Funct. Mater.* **2022**, 32, 2109542; *Nat. Commun.* **2022**, 13, 4805; *ACS Appl. Mater. Interfaces* **2022**, 14, 4571-4578) do not use standardized scratch resistance tests (e.g., friction coefficient), we compare the scratch resistance performance here (Figs. R2-R4) only based on the scratch morphology and the optical property changes before and after scratching. We can conclude that although all these optical coatings have good scratch resistance (e.g., the morphology and optical properties remain essentially unchanged after scratching), the standardized scratch tests that we used is apparently more violent and challenging, which side by side demonstrates the excellent scratch resistance of AMTA.

Figs. R2-R4:

Fig. R2. Scratch resistance tests of the photonic film (PDMS/BN), cited from “*Adv. Funct. Mater.* **2022**, 32, 2109542”.

Editorial Note: Fig. R4 below reproduced with permission from Dong et al., Slippery Passive Radiative Cooling Supramolecular Siloxane Coatings, *ACS Appl. Mater. Interfaces*, **14**, (3) 4571-4578. Copyright 2022 American Chemical Society.

Fig. R3. Scratch resistance tests of the AACP coating (PFOTS/TiO₂), cited from “*Nat. Commun.* **2022**, *13*, 4805”.

Fig. R4. Scratch resistance tests of the Supramolecular Siloxane Coatings (uPDMS/Al₂O₃), cited from “*ACS Appl. Mater. Interfaces* **2022**, *14*, 4571-4578”.

3. Is this coating assembled on a substrate? How is this material supposed to be used in applications? On thicker panels?

Response: Thank you for the constructive suggestions. For PDRC materials, there exist two main application routes, or usage forms, i.e., film or coating. As we report in this work, AMTA can be easily assembled as a lightweight film with good optical properties as well as excellent mechanical strength and environmental durability via a “Solvent exchange-Reprotonation” processing strategy. Based on this thin film form, it can be directly applied to automotive covers, outdoor tents and other cooling scenarios that require high portability. Moreover, we can also assemble AMTA in the form of paint by brushing-coating, drop-coating or spray-coating, depending on the viscosity of the AMTA suspension, which can be precisely controlled by solvent dilution. In order to improve the adhesion of the AMTA coating to the substrate, an “adhesive layer” is usually brushed onto the substrate first, followed by the construction of the AMTA coating. The above method works perfectly for most substrates, including aluminium, glass, plastic and wood products, as shown in Supplementary Fig. 28. Therefore, according to different cooling scenarios, we can choose the appropriate assembly methods of AMTA, thus further expanding the application range of PDRC materials.

Supplementary Fig. 28:

Supplementary Fig. 28. Assembly of AMTA on different substrates. To enhance the adhesion of AMTA to the substrate, a layer of adhesive was first brush-coating to the substrate such as aluminium, glass, plastic, wood, etc. Subsequently, AMTA suspension with different viscosity could be assembled on the substrate by brush-coating, drop-coating or spray-coating according to the construction needs.

4. The section “Results” starts with a section that reads like an introduction. The results section should focus on results.

Response: Thank you very much for the kind suggestion. We have revised the beginning of the “Results” section to focus more on results and differentiate it from the “Introduction” section, specifically the AMTA design principles.

Page 3-4, “Design, fabrication and characterization”:

To meet the long-term cooling requirements of outdoor devices in harsh environments, high solar reflectivity, excellent mechanical properties and outstanding environmental durability of radiative coolers will be the primary consideration. For solar reflectivity, favorable strong scattering created by impedance mismatch (*Electronics* **2019**, 8, 1022) at multiple interfaces has developed as its classical design principle. Mica@TiO₂, a commercial pearlescent pigment, creatively combines exfoliated Mica microplatelet (Core) with uniformly distributed TiO₂ nanograins (Shell, Supplementary Fig. 3) to

achieve both high refractive index ($k > 2.6$) and environmental durability, making it a desirable 2D inorganic scatterer (Fig. 1b,c). Besides, functional groups such as Si-C, Si-O and Ti-O within Mica@TiO₂ can greatly improve the emissivity of radiative coolers at the atmospheric window. Likewise, since the importance of polymer networks for the molding and processing of PDRC films, ANFs with large specific area, ultra-high strength (~3.6 GPa) and heat-resistance (> 400 °C, *Adv. Funct. Mater.* **2020**, *30*, 2000186) are appropriately used as an organic matrix to fabricate high mechanical performance materials.

5. Fig. 2d: The authors show microcracks in the film, and they assume that these cracks appeared during the tensile test. The authors should show that it is indeed the case, and that the crack did not appear during the fabrication process.

Response: Thank you very much for the kind suggestion. We re-conducted a more rigorous SENT experiment to fully reflect the crack propagation behaviors during AMTA stretching, as shown in Fig. 2d and Supplementary Fig. 11. By observing the surface morphology of the notched samples before and after stretching (Supplementary Fig. 11b,c), it can be seen that no damage exists near the notch before stretching, while the major crack expands from the notch after stretching, which demonstrates that the microcracks in AMTA are indeed generated during the tensile test. In addition, the discussion on the crack propagation behaviors has been revised in the manuscript page 5-6 and also detailed in the comment 3 from the reviewer #1. Please check the relevant content in the above locations if needed.

Supplementary Fig. 11a-c:

Supplementary Fig. 11. Crack propagation behaviors of AMTA in single-edge notched tensile (SENT) tests. a) Schematic illustration of SENT experiments. **b,c)** FESEM images of notched AMTA before (b) and after (c) stretching, demonstrating that circuitous crack propagation occurred indeed after stretching the notch.

6. Finally the manuscript contains typos, odd grammatical forms and bad choice of

words. It should be carefully edited and proofread.

Response: Thank you very much for the kind suggestion. We have double-checked the English to polish our manuscript and corrected the odd grammatical forms, styling and typos found in our manuscript. The revised texts has been marked in red in manuscript (for clarity, certain minor errors in localized locations have not been marked in detail), please see if the revised version met the English presentation standard.

REVIEWERS' COMMENTS

Reviewer #1 (Remarks to the Author):

The authors have well addressed the reviewers' comments. There is no further comments before its publication.

Reviewer #2 (Remarks to the Author):

The authors addressed all my concerns in the previous version of the manuscript. I think the paper is suitable for publication.

Reviewer #3 (Remarks to the Author):

The authors have properly addressed my comments. In my opinion the manuscript should be accepted.

Point-by-Point Responses to Reviewers' Comments

Reviewer #1 (Remarks to the Author):

The authors have well addressed the reviewers' comments. There is no further comments before its publication.

Response: Thank you for your positive comments.

Reviewer #2 (Remarks to the Author):

The authors addressed all my concerns in the previous version of the manuscript. I think the paper is suitable for publication.

Response: Thank you for your positive comments.

Reviewer #3 (Remarks to the Author):

The authors have properly addressed my comments. In my opinion the manuscript should be accepted.

Response: Thank you for your positive comments.